# A Vital Role of High-Pressure Processing in the Gel Forming on New Healthy Egg Pudding through Texture, Microstructure, and Molecular Impacts

**DOI:** 10.3390/foods11172555

**Published:** 2022-08-24

**Authors:** Chattraya Ngamlerst, Pattaneeya Prangthip, Bootsrapa Leelawat, Supattra Supawong, Suteera Vatthanakul

**Affiliations:** 1Department of Food Science and Technology, Faculty of Science and Technology, Thammasat University, Klong Luang, Pathumthani 12121, Thailand; 2Department of Tropical Nutrition and Food Science, Faculty of Tropical Medicine, Mahidol University, Ratchathewi, Bangkok 10400, Thailand; 3Thammasat University Center of Excellence in Food Science and Innovation, Klong Luang, Pathumthani 12121, Thailand

**Keywords:** high pressure processing, egg white, protein structure, microstructure

## Abstract

High-pressure processing (HPP) can induce gelation of egg-white protein and improve physical and physicochemical properties of egg-white pudding. Interestingly, one step, including production and pasteurisation, is accomplished to produce a ready-to-eat snack. An ideal healthy snack in the elderly population consists of low-sugar and fat, high fibre and vitamin levels, is tasty, creamy-soft, and refreshing. Our strawberry-flavoured egg-white pudding contains high protein and fibre from inulin, zero fat, and a soft texture produced for various groups, from healthy to older people. After HPP at different high-pressure levels (450, 475, and 500 MPa) and different times (5, 10 and 15 min), this study investigated the physical quality and physicochemical properties of strawberry-flavoured egg-white pudding, such as texture, colour, syneresis, microstructure, secondary structure of protein, and microorganism growth. The results indicate increasing high-pressure levels, and/or holding time treatment caused significantly (*p* < 0.05) higher hardness values and lower syneresis, as well as surface hydrophobicity. Moreover, many micropores and thicker wall structures were clearly observed for increasing high-pressure levels. Furthermore, HPP altered the β-sheet and β-turns structure of strawberry-flavoured egg-white pudding. In conclusion, increasing high-pressure levels and/or holding time treatment at 450, 475, and 500 MPa for 5, 10, and 15 min affected the physical, physicochemical, and biochemical properties of strawberry-flavoured egg-white pudding.

## 1. Introduction

Nowadays, more than 15 million people globally die from non-communicable diseases, such as cardiovascular disease, diabetes, and chronic kidney disease (CKD) [1]. A major factor driving these diseases is an unhealthy diet. Many patients are restricted to certain kinds of food. For example, CKD patients have limited potassium intake, which is abundant in fruit and vegetables, and are advised to eat high biological value protein, such as egg [2].

Egg protein contains all essential amino acids and is an affordable source of nutrients. To reduce fatty acid and excess calories, egg-white is a better choice as the main ingredient in snack production. Strawberry is the most popular fruit in the confectionary industry because it tastes good, has a good aroma, and has an attractive colour. Further, it contains very low potassium, which is good for CKD patients [3]. Inulin is a functional fibre or prebiotic dietary fibre that can promote intestinal health [4] and has been used as a fat replacer, due to its ability to improve food texture and mouth feel [5]. Rique et al. (2022) reported that the elderly population enjoy healthy snacks with low-sugar and fat and/or snacks rich in fibre and vitamins [6]. They also like tasty, aromatic, creamy-soft, and fresh/refreshing snacks, such as fruit, pudding, and jelly.

The high-pressure processing (HPP) is a novel technology in the food industry that involves cold pasteurisation to maintain nutrient levels in food and inhibit microorganism growth [7]. HPP outcomes can alter protein conformation by introducing protein aggregation, protein denaturation, or gelation, and they may change protein appearance [8,9,10,11]. Many commercial food products treated with HPP widely use inactive microorganisms [7,12,13]. Further, some products use HPP as a pre-treatment step to improve food’s physical and/or chemical properties [14,15,16], and HPP treatment can also induce egg-white gelation. Razi et al. (2019) studied the rheological properties of albumin treated with HPP and found that egg albumin had a stronger structure at 400 MPa [10]. Other studies focusing on the effect of HPP on textural and colour properties of egg-white also reported higher values for hardness and turbidity parameters [17,18]. However, no reports using only HPP as a one-step procedure to produce ready-to-eat food have been conducted.

This study aimed to investigate and understand the effect of high-pressure levels and holding times on the physical, physicochemical, and biochemical properties of strawberry-flavoured egg-white pudding. This food item may be an optional healthy snack for some patients and people needing to prevent loss of appetite and malnutrition and consume high protein, high dietary fibre, and low fat snacks.

## 2. Materials and Methods

### 2.1. Materials

Egg-white (hen’s egg, pH 7.25, food grade, protein 86%) was a gift from DKSH (Bangkok, Thailand) Limited. Inulin (Fuji FF inulin, degree of polymerisation > 14) was obtained from Fuji Nihon Thai Inulin Company, Limited (Bangkok, Thailand). Strawberry concentrate (strawberry squash, 56.7°Brix, pH 2.62) was purchased from Doi Kham Food Products Co., Ltd., Bangkok, Thailand. Citric acid and malic acid were purchased from Krungthepchemi Co., Ltd., Bangkok, Thailand. Ellman’s reagent (5,5′-Dithiobis (2-nitrobenzoic acid)) was purchased from Merck Ltd. (Darmstadt, Germany). The 8-Anilino-1-naphthalenesulfonic acid (fluorescence probe) was purchased from Tokyo Chemical Industry Co., Ltd. (Tokyo, Japan).

### 2.2. Sample Preparation

Egg-white powder was mixed homogeneously with drinking water (1:3.5, *w*/*w*), under magnetic stirring, at 25 °C. Egg-white solution was then mixed with sugar, citric acid powder, malic acid powder, inulin powder, and strawberry concentrate (0.21:0.008:0.008:0.41 1:1.76, *w*/*w*). The mixture of liquid egg-white and strawberry juice (165 g each) was then transferred to a polypropylene (PP) cup (7.5 cm diameter, 5.7 cm base, and 5.5 cm height) with polyethylene terephthalate (PET)/cast polypropylene (CPP) top film sealing (52 microns). High-pressure treatments were performed in a 5-litre chamber high-pressure machine (BaoTou KeFa high pressure technology Co., Ltd., BaoTou, China). Samples were introduced into the chamber and treated with 450, 475, and 500 MPa for 5, 10, and 15 min at 25 °C. Following high-pressure pasteurisation, samples were maintained in chilled room, at 4 °C for 24 h, for further analysis.

### 2.3. Texture Analyses

Hardness and springiness of strawberry-flavoured egg-white pudding were measured using a TA-XT2i texture analyser (Stable Micro System Ltd., Surrey, UK). The protocol was adapted from Leemud et al. [19]. Pudding placed in a cylinder-shaped PP cup was left at room temperature (25 °C) for 30 min. Hardness was measured using a compression test mode with a measurement speed of 1.0 mm/s, with 50% strain penetration depth using a hemispherical probe (P/0.5HS). The trigger force was set to 5 gf. Pre- and post-speed were set at 1.0 and 10 mm/s, respectively, and the data acquisition rate was 200 pps. Absolute positive force was calculated and defined as hardness. Pudding springiness was measured at a speed of 1.0 mm/s with 10% strain penetration depth, using a hold until time test mode with a hemispherical probe (P/1R), and held for 60 s. Springiness was calculated using the following equation:Springiness = Force at 60 s/Maximum force

### 2.4. Colour Measurement

The surface colour of pudding samples was measured for redness (a*, red-green coordinate) and yellowness (b*, yellow-blue coordinate), by the colour spectrophotometer (Hunter Lab CX 2687, Reston, VA, USA), and expressed as hue angle (H°). Hue angle was calculated by converting a* and b* using the following equation:$$H^{\circ }=\arctan\;(\frac{b^{*}}{a^{*}})$$

### 2.5. Syneresis Measurements

Syneresis measurements were performed, following’s method [20], with some modification. Strawberry-flavoured egg-white pudding samples were removed from the container and weighed. They were placed upside down on an 80 mesh strainer and drained for 15 days at 4 °C. Drainage was weighed and recorded every 3 days.

### 2.6. Surface Hydrophobicity

Surface hydrophobicity of strawberry-flavoured egg-white pudding was determined using Gao’s method [21] with some modification. Freeze-dried samples (1 g) were mixed well with 9 mL of phosphate buffer (0.01 M, pH 7.4). Solutions were centrifuged, at 10,000× *g* at 4 °C, for 15 min. Supernatants were then diluted to 0.05, 0.1, 0.2, 0.4, and 0.8 mg/mL with the same phosphate buffer. Samples were then filtered through a polyethersulfone membrane with a 0.45 µm pore diameter. Then, 4 mL of each sample were mixed with 20 µL of ANS solution (fluorescence probe) and kept in the dark for 15 min. Fluorescence intensities (FI) were measured using a Synergy H1Hybrid fluorescence spectrometer (BioTek, Santa Clara, CA, USA), with an excitation wavelength at 395 nm and emission at 475 nm. An index of surface hydrophobicity (S_0_) was calculated from the FI slope plotted against the five different sample concentrations.

### 2.7. Free Sulfhydryl Group Determination

The free sulfhydryl group of egg-white pudding was measured, according to Luo et al. [22], with some modification. Freeze-dried egg-white pudding powder (20 mg) was mixed with 4 mL of 2.5% SDS–TGE buffer (10.4 g Tris-6.9 g glycine-1.2 g EDTA per litre, defined as TGE pH = 8.0). Samples were then incubated with a nutating shaker at room temperature for 60 min, followed by centrifugation at 12,000× *g* for 10 min. The supernatant was then collected and adjusted to a total volume of 5 mL using the SDS–TGE buffer. The sample was then mixed with 40 µL of Ellman’s reagent and incubated for 30 min at ambient temperature. Supernatant absorbance was then measured at 412 nm. An extinction coefficient of 13,600 M^−1^ cm^−1^ was applied for calculations.

### 2.8. Fourier Transform Infrared Spectroscopic (FT-IR) Measurements

Freeze-dried strawberry egg-white pudding samples were carefully placed on an attenuated total reflectance device. Mid-infrared spectra of samples were detected using a FT-IR spectrometer (Nicolet iS50, Thermo Scientific, Waltham, MA, USA). The scan spectra, at 4 cm^−1^ resolution for 64 scans, were averaged in a 400–4000 cm^−1^ region, and the spectra were analysed using OMNIC software. Secondary derivations were determined by OriginPro2021 (OriginLab Corporation, Northampton, MA, USA) following the method of Sadat and Joye [23]. The Levenberg–Marquardt algorithm was used for non-linear fitting of peaks in the spectral data at the amide I region, which had a wavenumber between 1600 and 1700 cm^−1^. Baseline corrections were performed using the second derivative (zeros) method. The Voigt function was used to perform peak fitting. α-helix, β-sheet, turns, and random coil content were calculated by dividing the areas under the bands by the total secondary structure area under the amide I band.

### 2.9. Microstructure

Field Emission Scanning Electron Microscope (FE-SEM) (model JEOL JSM7800F, Tokyo, Japan) was used to observe microstructures of freeze-dried strawberry-flavoured egg-white pudding. All samples were coated with gold particles and observed using 200× magnification at a 2 kV acceleration voltage. Samples were randomised by blinding analysis without bias from the Center of Scientific Equipment for Advanced Research, Thammasat University.

### 2.10. Microbiological Analysis

Each sample, in a sealed cup, was stored in a cold room (4 °C) for 15 days. Bacterial culture mediums were prepared, including Trypticase Soy Agar (TSA), Potato Dextrose Agar (PDA), TSA + 7.5% NaCl, Eosin Methylene Blue agar, and Oxford agar. Microbial growth was tested every 3 days. Ten grams of each sample, in duplicate, were mixed with peptone water and serially diluted with the same solution. Dilutions were then spread on a specific media, after which testing plates were incubated at 37 °C for 24 h. Total plate count, yeast-mould (incubated at room temperature for 72 h), *Bacillus cereus*, *Staphylococcus aureus*, *Escherichia coli*, *Salmonella* spp., and *Listeria monocytogenes* were then evaluated after incubation.

### 2.11. Statistical Analysis

The 3 × 3 factorial in a Complete Randomised Design experimental design was used in this study (three levels of high-pressure processing × 3 levels of time). All assays were performed at least three times and each time used three technical replicates. Statistical analysis was performed using the SPSS software package for the Windows trial version (IBM^®^ SPSS^®^ Statistics, Armonk, NY, USA). Analysis of variance was used to compare results using the Duncan’s multiple range test with a confidence interval of 95% (significant difference was detected if *p* < 0.05).

## 3. Results and Discussion

### 3.1. Texture and Colour Analysis

Food texture is a key food attribute, especially in a semi-solid food such as pudding. Hardness and springiness play an important role in the acceptability of consumers and quality control for production. High-pressure levels and holding times on hardness and springiness levels of strawberry-flavoured egg-white pudding are shown in Table 1. The highest hardness value was observed in pudding treated with 500 MPa for 15 min, while the lowest was found in pudding treated with 450 MPa for 5 min. Treatment prepared with high-pressure levels at 500 MPa, most likely, showed a significantly (*p* < 0.05) higher hardness value than 475 and 450 MPa, due to changes in protein structure, from no pore at the surface area of pudding to sponge-like, which clearly showed in the microstructure observation and formation of a stronger gel network. Kapoor et al. (2021) used high-pressure levels, at 500 MPa, for 15 min on cottage cheese to reveal high hardness values. Further, pudding hardness value increased significantly (*p* < 0.05) with increasing holding times. For the impact of holding time on pudding texture, the average hardness value after 15 min was higher than 10 and 5 min, respectively. The appearance of strawberry-flavoured egg-white pudding became stronger with increasing holding time, as clearly observed at 450 MPa (Figure 1). Gel structure was good for building and had a more solid appearance, which was related to hardness value in Table 1. Using the lowest high-pressure level or shortest holding time caused the highest springiness value, which were 0.561 and 0.538, respectively (*p* < 0.05). Conversely, using highest high-pressure level or the longest holding time caused significantly (*p* < 0.05) lower springiness values, which were 0.431 and 0.455, respectively. These results indicate strawberry-flavoured egg-white pudding has a soft texture, but it is not elastic when treated with higher pressure levels over long periods.

The average of hue angle of the pudding, treated with 450 MPa, had the lowest value, which was 63.50°. According to a colour wheel subtends of 360° [24], which represent red-purple to yellow at an angle beginning at 0° and moving to 90°, respectively, those values were significantly shaded redder (*p* < 0.05) than the pudding treated with 475 MPa (65.00°). The pudding treated with 500 MPa had the highest hue angle (66.54°), which was significantly shaded more yellow (*p* < 0.05) than the pudding treated with 450 MPa. However, using three different holding time levels were not significant changes in the hue angle.

### 3.2. Syneresis

Liquid separation from the pudding to the surface is called syneresis and represents quality change in pudding products [25]. Syneresis percentage of egg-white pudding significantly (*p* < 0.05) improved under increasing high-pressure levels and/or holding time. Total syneresis percentage, during 15 days of chilled storage, ranged from 29.61% to 40.08% (Table 1). Pudding treated with 450 MPa for 5 min showed the highest syneresis percentage, which may be due to high-pressure levels at 450 MPa, caused by weak gel, and the inability to form a strong three-dimensional network structure, as verified from microstructure observations. Another possible explanation for high syneresis may be low pH levels, which was 4.33 and far from the isoelectric point (4.5) of albumin [26]. Therefore, the increasing positive charge of protein led to fewer binding sites for water.

However, results were insufficient to measure water release from pudding treated with 500 MPa after day 8 or after a 15 min holding time (Figure 2). This could be due to higher pressure and/or a longer holding time due to proteins unfolding as a result of high exposure of the hydrophilic group, where there are more water binding sites [27].

### 3.3. Determining Surface Hydrophobicity

Changes in fluorescence intensity were represented by protein unfolding and the exposed hydrophobic core to the protein molecule surface. The effect of high-pressure treatment on relative protein surface hydrophobicity (S_0_) of egg-white pudding is shown in Figure 3. The S_0_ value significantly decreased with increasing high-pressure from 450 to 500 MPa (*p* < 0.05). Holding time during high-pressure treatment also affects the S_0_ value, whereby increasing holding time from 5 to 15 min significantly decreased the S_0_ value (*p* < 0.05). The highest S_0_ value was achieved at 450 MPa for 5 min, which was almost three-fold higher than 500 MPa for 5 min. This decreasing S_0_ value may be explained by very high pressure (>400 MPa), leading to protein structure refolding, and hiding some of the hydrophobic components. These findings corroborate with a study reported a decreasing S_0_ value of ovaltransferrin treated with increasing high-pressure from 400 MPa to 500 MPa [8]. Similar results were reported a significant decrease in surface hydrophobicity when high-pressure levels changed from 400 MPa to 500 MPa on rice bran proteins [27]. Previous study reported a relative surface hydrophobicity of high-pressure treated (100–500 MPa) egg yolk, which significantly decreased with increasing pressure levels. This study indicated that higher pressure levels caused refolding, protein aggregation, and formed a more stable structure that reflected hardness values [28].

### 3.4. Free Sulfhydryl Group

Egg-white protein contains four free-SH groups buried in the core protein structure [29]. After high-pressure treatment (450–500 MPa), our results indicate that free-SH content showed no significant change at 5–15 min of holding time at 475 and 500 MPa (Figure 4). However, free-SH content significantly increased (*p* < 0.05) for longer holding times at 475 MPa. Results showed the highest free-SH content in samples treated with 450 MPa, and they were significantly higher (*p* < 0.05) than samples treated with 500 MPa. Moreover, the pudding treated with 450 MPa for 15 min showed the lower free-SH content, which related to the lower protein surface hydrophobicity as well (Figure 4). These outcomes were related to relative protein surface hydrophobicity in our study, which showed a similar trend. Since S_0_ values in our study indicated refolding and rearrangement of the protein structure, decreasing free-SH content formed a stronger protein structure of intermolecular sulfhydryl-disulfide and/or disulfide bonds [17].

### 3.5. FT-IR Measurements and Secondary-Structure Analysis

We used Fourier transform infrared spectroscopy to study the transition of protein conformations. The FT-IR spectra of strawberry-flavoured egg-white pudding are shown in Figure 5. The amide I region, at wavenumber 1600–1700 cm^−1^, represented the secondary structure of protein. β-sheet was identified at the FT-IR spectra at 1600–1639 cm^−1^. The subpeak of 1640–1650 cm^−1^ represented a random coil or disordered conformation. The peak, ranging from 1651 to 1660 cm^−1^, was attributed to the α-helix. Lastly, the peak at 1661–1700 cm^−1^ was assigned to β-turns [30,31,32]. The only peak in the amide I region ranged between 1639.99 and 1644.49 cm^−1^, which related to the β-sheet and random coil structures. All treatments showed a peak between 1639.99 and 1640.46 cm^−1^, except pudding treated with 475 MPa for 5 min and 500 MPa for 5 and 10 min, which showed peaks between 1644.09 and 1644.49 cm^−1^. These results indicated that peaks tend to slightly shift to a higher wavenumber when compared to high-pressure treatment at 450 MPa. Curve fitting, which was calculated from peak areas, is shown in Figure 6(1). Higher pressure levels caused a structural transition, from a β-sheet to β-turns structure, under 5 min holding time in HPP. Conversely, using 10 min and 15 min holding times promoted a structural transition, from a β-turns to β-sheet structure, under increasing high-pressure levels. In addition, our study indicated that high-pressure levels and holding times did not cause any change in α-helix and random coil structures. A change in a β-turns structure to a β-sheet structure was related to texture properties, which showed high hardness and low springiness values of the egg-white pudding. This outcome may be due to increasing β-sheet levels, leading to higher intermolecular hydrogen bonds that cause more protein–protein interactions and protein aggregation [33,34].

### 3.6. Microstructure Observation

We investigated the microstructure of egg-white pudding after HPP using a field mission scanning electron microscope with 200× magnification, as shown in Figure 6(2a–2i) The egg-white pudding, treated with high-pressure at 450 MPa for 5–10 min (Figure 6(2a,2b)), showed almost no pores and a very dense, homogeneous structure most likely from no water retention in the network gel structure. However, pudding treated with high-pressure, at 450 MPa for 15 min (Figure 6(2b,2c)), showed large pores. Microscopic observations showed close relationships to water binding capacity and textural properties. This could explain a higher percentage of water syneresis and lower hardness value compared with higher HPP levels. Further, pudding treated with high-pressure, at 475 MPa for 5–15 min (Figure 6(2b,2d–2f)), showed a sponge-like appearance, and the surface revealed many tiny pores but with a thin porous wall. Under a holding time of 5 min at 475 MPa, the network of micropores became a single large pore (Figure 6(2b,2d)). After HPP at 500 MPa, for 5–15 min (Figure 6(2b,2g–2i)), the pudding showed a spongy structure with larger pores and a more open structure compared with HPP at 475 MPa. Large pores with a very thick wall may be a result of establishing a strong protein network and low water syneresis. Holding time for 15 min, at 500 MPa, caused smaller pores and presented a morphology of a thicker network structure compared with a holding time of 5–10 min. Microstructure observations were consistent with increasing hardness values and synesis of egg-white pudding. These findings also agree with researcher who observed that HPP treatment of cold smoke salmon, at 400–500 MPa, increased the space among myofibril cells [35]. They also found increasing pressure levels and longer processing times caused larger spaces and led to higher hardness values. Xue et al. (2017) also found that a rabbit meat sausage showed a more interconnected network under an HPP greater than 200 MPa [36].

### 3.7. Microbiological Analysis

After 15 days of storage at 4 °C, total plate count, yeast mould, *Bacillus cereus*, *Staphylococcus aureus*, *Escherichia coli*, *Salmonella* spp., and *Listeria monocytogenes* remained undetected in all pudding samples (data are not shown). HPP is a cold pasteurisation technique that destroys some bacteria and prevents spore germination. Moreover, strawberry-flavoured egg-white pudding, in this study, is an acidic food (pH 4.3), which could show improved product shelf life for at least 15 days at 4 °C.

## 4. Conclusions

HPP treatment at 450, 475, and 500 MPa for 5, 10, and 15 min altered the conformation of egg-white protein. Increasing high-pressure levels and/or holding time treatment led to higher hardness values, as well as lower syneresis and surface hydrophobicity, which may be due to changes in the secondary structure of egg-white protein and protein unfolding. Increasing high-pressure levels from 450 to 500 MPa led to five times higher hardness values and 25.4% lower syneresis. We also found partial reversible protein refolding and rearrangement. Our results were consistent with the reduced free-SH groups. Different high-pressure levels and treatment durations can shift β-sheet to a β-turns structure, but they cannot induce any change in α-helix and random coil structures. Microstructure observations indicate that HPP, at higher levels, caused many micro pores and presented a thicker wall structure morphology. Post-package pasteurisation of strawberry-flavoured egg-white pudding by HPP, in this study, has potential as a safe food with good texture. Moreover, strawberry-flavoured egg-white pudding may be a healthier choice for the elderly population. Sensory evaluations may be needed to further explore sensory quality and consumer preference.

## Figures and Tables

**Figure 1 foods-11-02555-f001:**
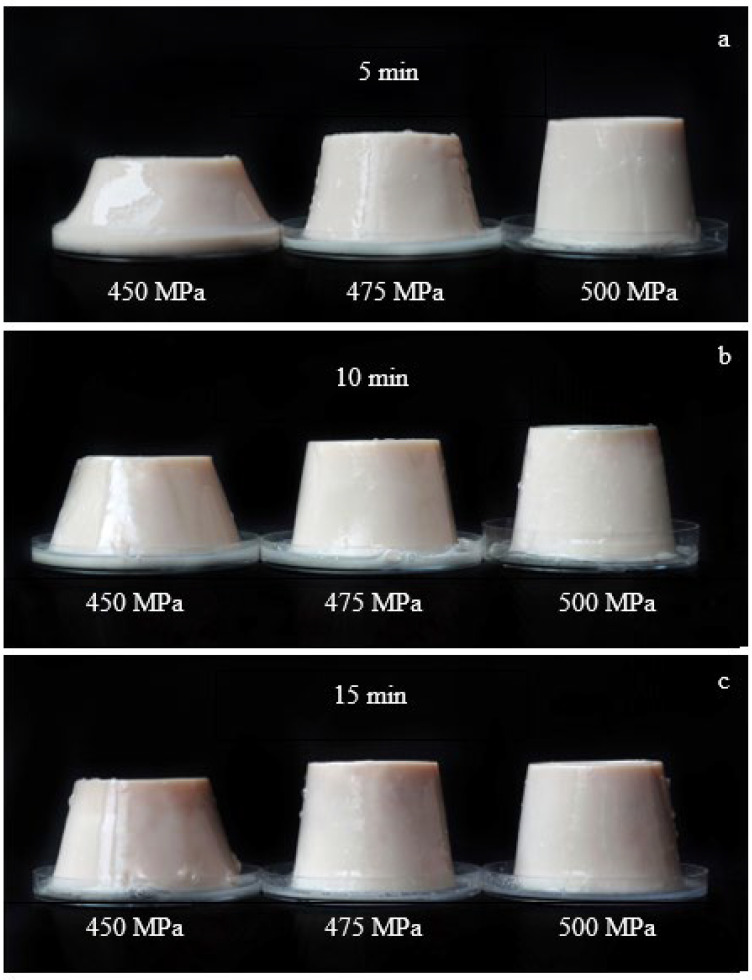
Effect of high-pressure on appearance of strawberry-flavoured egg-white pudding for different holding times: 5 min (**a**), 10 min (**b**), and 15 min (**c**).

**Figure 2 foods-11-02555-f002:**
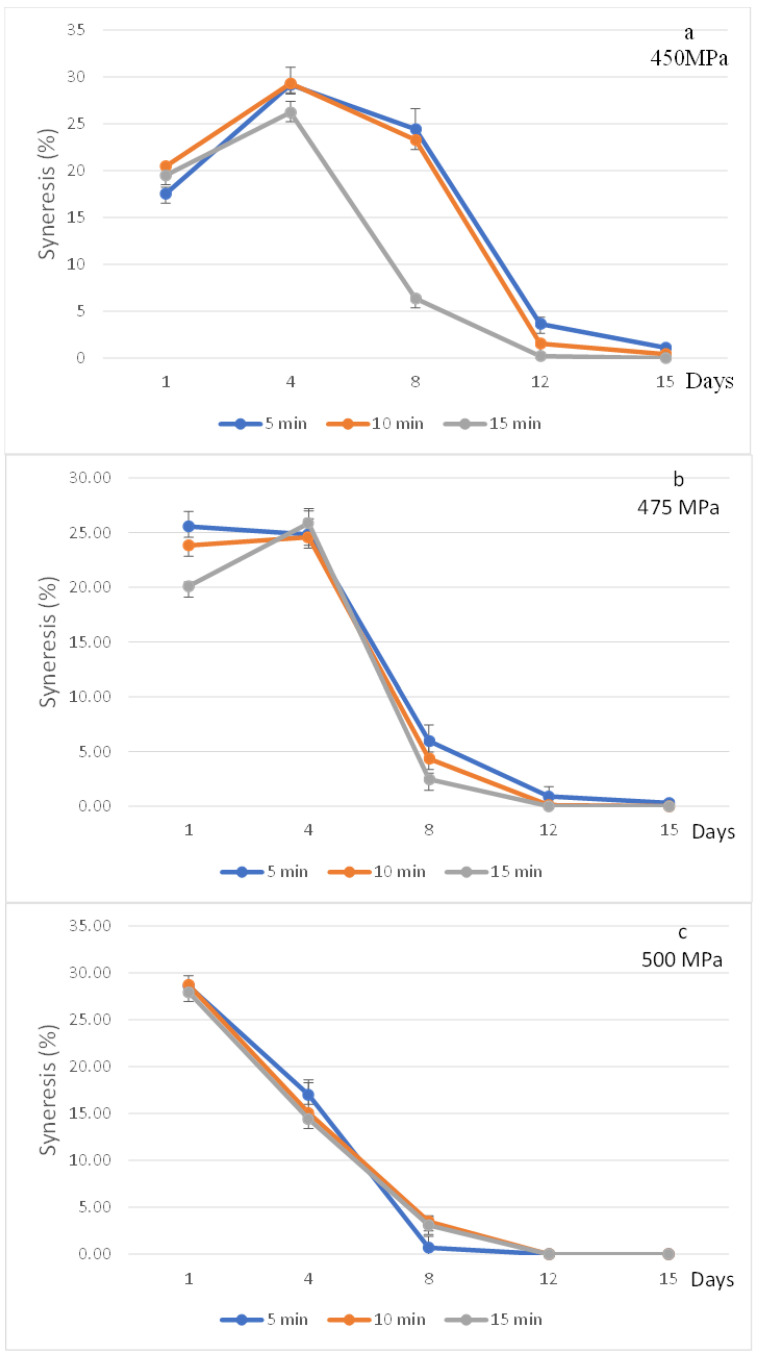
Effect of high-pressure and time on syneresis percentage of strawberry-flavoured egg-white pudding during 15 days storage: 450 MPa (**a**), 475 MPa (**b**), and 500 MPa (**c**).

**Figure 3 foods-11-02555-f003:**
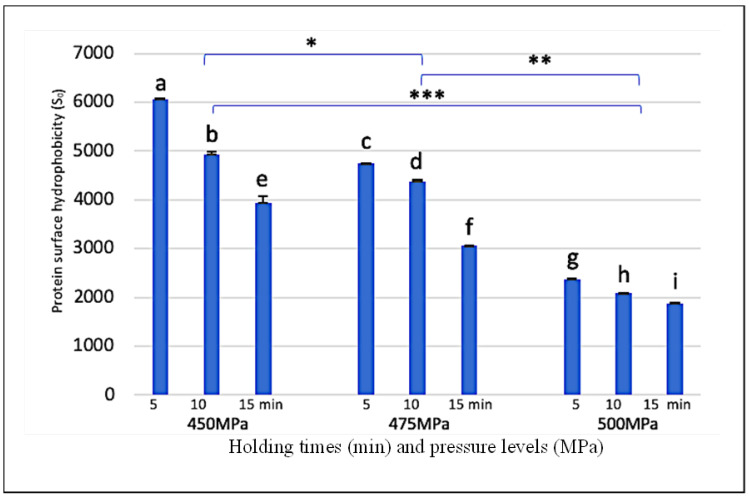
Effect of high-pressure and time on surface hydrophobicity of strawberry-flavoured egg-white pudding. Different lowercase letters are significant different (*p* < 0.05). Different stars (*,**,***) indicate significant differences (*p* < 0.05) between group of high pressure levels.

**Figure 4 foods-11-02555-f004:**
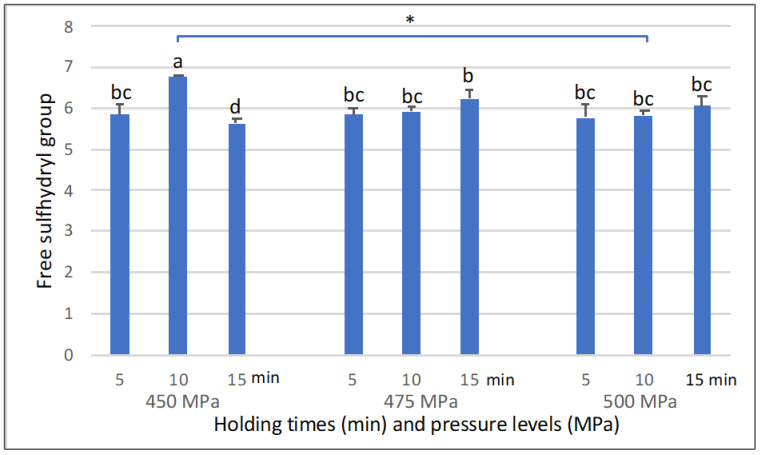
Effect of high-pressure and time on free sulfhydryl group content of strawberry-flavoured egg-white pudding. Different lowercase letters are significant different (*p* < 0.05). * indicates significant differences (*p* < 0.05) between group of high-pressure levels.

**Figure 5 foods-11-02555-f005:**
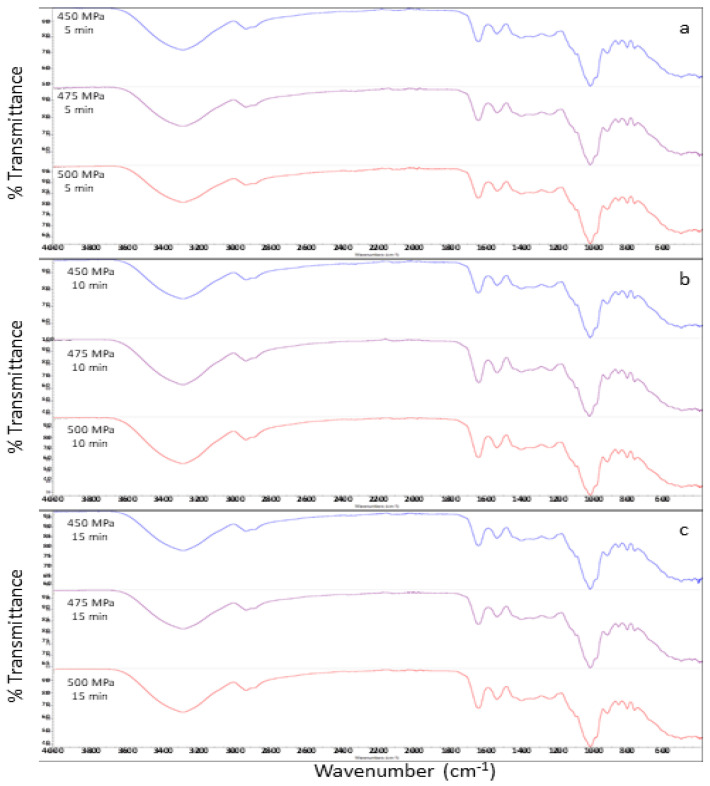
Effect of high-pressure and time on the FTIR spectra of strawberry-flavoured egg-white pudding. (**a**–**c**) is FTIR stack spectra of treated with 450–500 MPa for 5, 10, 15 min, respectively.

**Figure 6 foods-11-02555-f006:**
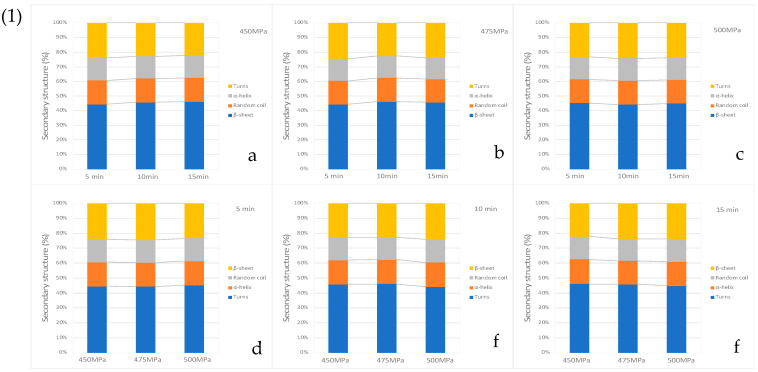
Effect of high-pressure and time on (**1**) the percentage of secondary: 450 MPa at 5, 10 and 15 min (**a**), 475 MPa at 5, 10 and 15 min (**b**), 500 MPa at 5, 10 and 15 min (**c**), 5 min at 450, 475 and 500 MPa (**d**), 10 min at 450, 475 and 500 MPa (**e**), 15 min at 450, 475 and 500 (**f**) and (**2**) the microstructure of protein of strawberry-flavoured egg-white pudding with 200× magnification and 100 µm scale bar: 450 MPa 5 min (**a**), 450 MPa 10 min (**b**), 450 MPa 15 min (**c**), 475 MPa 5 min (**d**), 475 MPa 10 min (**e**), 475 MPa 15 min (**f**), 500 MPa 5 min (**g**), 500 MPa 10 min (**h**), and 500 MPa 15 min (**i**).

**Table 1 foods-11-02555-t001:** The effect of different high-pressure levels and holding times on physical properties of strawberry-flavoured egg-white pudding.

Samples	Hardness (gf)	Springiness (-)	Hue Angle (H°)	Total Syneresis (%)
A: Pressure level				
450 MPa	43.68 ^c^	0.561 ^a^	63.50 ^c^	38.46 ^a^
475 MPa	128.22 ^b^	0.470 ^b^	65.00 ^b^	34.85 ^b^
500 MPa	205.24 ^a^	0.431 ^c^	66.54 ^a^	30.67 ^c^
SL	0.00	0.00	0.00	0.00
B: Time				
5 min	78.31 ^c^	0.538 ^a^	64.73	36.79 ^a^
10 min	142.73 ^b^	0.469 ^b^	64.88	34.92 ^b^
15 min	156.12 ^a^	0.455 ^c^	65.43	32.26 ^c^
SL	0.00	0.00	0.126	0.00
Interaction A × B				
SL	0.00	0.00	0.002	0.00
Samples (pressure (MPa) /time (min.))				
450/5	10.04 ^h^	0.686 ^a^	63.25 ^ef^	40.08 ^a^
450/10	63.23 ^g^	0.514 ^b^	62.79 ^f^	39.78 ^a^
450/15	57.78 ^g^	0.482 ^c^	64.45 ^cde^	35.52 ^b^
475/5	90.99 ^f^	0.483 ^c^	65.67 ^d^	38.47 ^a^
475/10	142.63 ^d^	0.477 ^c^	64.28 ^de^	34.41 ^b^
475/15	151.05 ^c^	0.451 ^d^	65.07 ^cd^	31.66 ^c^
500/5	133.89 ^e^	0.446 ^d^	65.28 ^cd^	31.83 ^c^
500/10	222.31 ^b^	0.416 ^f^	67.58 ^a^	30.57 ^cd^
500/15	259.52 ^a^	0.431 ^e^	66.77 ^ab^	29.61 ^d^
Duncan	0.00	0.00	0.00	0.00

Note: Data are expressed as means. Different letters in the same column indicate significant differences (*p* < 0.05). SL: significant level; NS: not significant.

## Data Availability

Not applicable.

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
