# Peer review of "A Vital Role of High-Pressure Processing in the Gel Forming on New Healthy Egg Pudding through Texture, Microstructure, and Molecular Impacts"

_foods, 2022, doi:10.3390/foods11172555_

Round 1
Reviewer 1 Report
Dear author,
This paper is good. However, need some improvement. Kindly refer attached file for revision.
Thank you

Author Response
According to comments, we agreed and corrected with all the comments raised by reviewers. We would like to take this opportunity to express our sincere thanks to reviewer who identified areas of our manuscript that needed corrections or modification. We would like also to thank you for allowing us to resubmit a revised copy of the manuscript. Please see the file attached for revision.
|
Reviewer 1’s Comments |
Author’s Response |
|
Why 450-500 MPa were chosen in this study.
|
According to the preliminary study using 100-600 MPa (data was not showed), the strawberry egg white pudding started to form a gel like pudding at 450 MPa for 5 min. The pudding’s texture was too hard when using high-pressure at 500 MPa for 15 min. Therefore, high-pressure levels at 450-500 MPa were chosen in this study.
|
|
Title: The research title should be clear. What is healthy referring to?
|
From our introduction section, we already mention about the benefits of egg pudding. It might be a new alternative snack choice for CKD patients and elderly population. Thus, the word “healthy” is choose.
|
|
Abstract: Please include a brief conclusion in the abstract.
|
As suggested, we added a brief conclusion in the abstracts as following “ In conclusion, increasing high-pressure levels and/or holding time treatment at 450, 475 and 500 MPa for 5, 10 and 15 min affected the physical, physicochemical and biochemical properties of strawberry-flavoured egg-white pudding”.
|
|
Line 195 – 202: This statement needs a reference.
|
We did add the reference of Mizrahi, S. (2010). Syneresis in food gels and its implications for food quality. Chemical Deterioration and Physical Instability of Food and Beverages, 324–348.
|
|
Line 202 – 203: This statement needs a reference.
|
We did add the reference of He, W., Xiao, N., Zhao, Y., Yao, Y., Xu, M., Du, H., … Tu, Y. (2021). Effect of polysaccharides on the functional properties of egg white protein: A review. Journal of Food Science, 86(3), 656–666.
|
|
Table 1: No discussion reported for hue angle.
|
We thanks for your helpful suggestion. We have added our hue angle data in the result.
|
|
Line 241: (400 – 500 MPa) - is this correct?
|
Corrected all as suggested. We changed to (450–500 MPa) |
|
Line 261: Milošević et. Al. – is this correct?
|
Corrected all as suggested. We changed to Milošević et. al., |
|
Line 262: 1639.99 and1644.49 – a space is missing.
|
Corrected all as suggested. We added the space of 1644.49 cm−1 |
|
Line 277: A punctuation is missing.
|
Corrected all as suggested. We added a punctuation ; |
|
Line 312 – 317: What is the shelf life of egg pudding at 4°C?
|
Thank you very much to reviewer to raise this point. We did not measure the shelf life of egg pudding at 4 °C. Normally, the shelf life of pasteurization of low acid food product (such as milk) at 4°C is around 7-10 days for maintaining best quality and taste. Our pudding is acid food (pH 4.3). At 4°C for 15 days, pathogen bacteria, total plate count, and mold & yeast were not detected in our product. Therefore, from our result, we could say only our egg’s shelf life is more than 15 days.
|

Reviewer 2 Report
This study evaluated the effect of high-pressure processing in gel forming on egg pudding through texture, microstructure and molecular impacts using different techniques. The subject of this study is interesting, and results are novel and promise. However, it needs revision to be accepted for publication. The effect of HPP on the nutritional composition was not determined in this study. No explanation included why 450 – 500 MPa were chosen in this study. Other comments are listed below.
Title: The research title should be clear. What is healthy referring to?
Abstract: Please include a brief conclusion in the abstract.
Line 195 – 202: This statement needs a reference.
Line 202 – 203: This statement needs a reference.
Table 1: No discussion reported for hue angle.
Line 241: (400 – 500 MPa) - is this correct?
Line 261: Milošević et. Al. – is this correct?
Line 262: 1639.99 and1644.49 – a space is missing.
Line 277: A punctuation is missing.
Line 312 – 317: What is the shelf life of egg pudding at 4°C?
Author Response
According to comments, we agreed and corrected with all the comments raised by reviewers. We would like to take this opportunity to express our sincere thanks to the editors and reviewers who identified areas of our manuscript that needed corrections or modification. We would like also to thank you for allowing us to resubmit a revised copy of the manuscript. Please see the file attached for revision.
|
Reviewer 2’s Comments |
Author’s Response |
|
Line 36 – check the citation format (Mahan) |
Corrected all as suggested. We changed the citation format to (Mahan and Raymond, 2017). |
|
Line 50 – The HPP |
Corrected all as suggested. We added The |
|
Line 89 – samples were maintained at 4 deg in where? Need to state clearly
|
The puddings were maintained in chilled room at 4ºC. |
|
Line 178 – changes in protein structure need to be explained in detail related to the pressure exposed to the pudding during treatment |
Explained more as suggested to “Changes in protein structure from no pore at surface area of pudding to spongey like which clearly showed in microstructure observation (Fig. 6b) ” |
|
Line 184 – Pudding became weaker due to holding time need the scientific theory here |
We thank you for your kind concern. We added the scientific theory of Gel structure was good building and more solid appearance which was related to hardness value in table 1. |
|
Figure 2 – need x-axis label |
Corrected all as suggested. |
|
Line 231 – Yan et al. (2010) considered outdated and more than 10 years of citation. The author needs to update this. Replace the citation in lines 241, 249 and the rest, please. |
Thank you for your kind consideration. We update citation as you suggested. We still leave some references that their knowledge is still update as shown more citation with the no new knowledge replaced. |
|
Need details about data in Figure 4. Free-SH groups at 15 mins, 450 MPa as shows the lowest and please explain the relationship with Figure 3 within the same treatment. |
Added data in Figure 4 as suggested . The relationship of figure 4 and 3 are explained following “Moreover, the pudding treated with 450 MPa for 15 min showed the lower free-SH content which related to the lower protein surface hydrophobicity as well (Fig.4). These outcomes were related to relative protein surface hydrophobicity in our study, which showed a similar trend. Since S0 values in our study indicated refolding and rearrangement of the protein structure, decreasing free-SH content formed a stronger protein structure of intermolecular sulfhydryl-disulfide and/or disulfide bonds (Plancken et al., 2007).”
|
|
Figure 4 x-axis label? |
Corrected as suggested in figure 4. |
|
Check et al., format carefully for all text, eg. line 261 Milošević et. Al., 2021, Ji et al., 2015 supposedly |
Corrected as suggested. |
|
Figure 5 X & Y -axis labelling is too small and hardly seen |
Corrected as suggested in figure 5. |
|
Line 277 - Abrosimova et. al., 2016; Uygun-Saribay et al., 2017 |
Corrected as suggested. |
|
All figures in 6a. no X-axis labels and too small labels for the percentage, the SEM labels were tiny. Need to expose the magnifications. |
Corrected as suggested in figure 6. |
|
The comparison samples in lines 299- 304 were solid. Need more close comparisons possibly from semi-solid samples within the same treatment. |
Thank you for your kind consideration. The researches about using HPP higher than 450 MPa in semi-solid food was not found in a decade. However, sausage and salmon are high protein food similarly to egg white in our study. Those studies used a HPP level quite close to our study. Therefore, we compared our samples to those studies. |
|
Line 311-317 – need to discuss clearly how HPP affected the microbiological results. Any data/graph for this part? |
We discussed that “HPP is a cold pasteurisation technique that destroys some bacteria and prevents spore germination. Moreover, strawberry-flavoured egg-white pudding in this study is an acidic food (pH 4.3), which could show improved product shelf life for at least 15 days at 4ºC.” Since bacteria pathogens were undetectable and there are many figures in our results, we are sorry to not proposed it in our manuscript. |
|
Conclusion – change to conclusions, need details here eg. %, etc. to conclude the trends. |
Corrected as suggested. Details of “ Increasing high-pressure levels from 450 to 500 MPa led to 5 times higher in hardness values and 25.4% lower in syneresis.” are added in conclusions. |
|
Some of the grammatical errors were found and authors need to thoroughly check.
|
Thoroughly check as suggested by Enago, publication support and language services. |
|
Line: |
Corrected all as suggested |
